# Inclusive Socialization? The Relationships between Parents’ and Their Child’s Attitudes toward Students with Disabilities

**DOI:** 10.3390/ijerph192416387

**Published:** 2022-12-07

**Authors:** Sara Santilli, Maria Cristina Ginevra, Moshe Israelashvili, Laura Nota

**Affiliations:** 1Department of Philosophy, Sociology, Education, and Applied Psychology, University of Padova, 35131 Padova, Italy; 2School of Education, Tel Aviv University, Tel Aviv-Yafo P.O. Box 39040, Israel

**Keywords:** parents’ attitudes, children’s attitudes, disability, school inclusion

## Abstract

Promoting social and school inclusion of children with disabilities has been associated with their health-related quality of life. This study aimed to analyze the connection between parents and children’s attitudes toward disabilities as one factor contributing to the inclusion and well-being of individuals with disabilities. Three types of disabilities—i.e., attitudes toward a child with a sensory disability (hearing), a child with an intellectual disability (Down syndrome), and a child with problems with aggressiveness and angry outbursts (behavioral problems)—were examined. A sample of 598 White Italian elementary school students (303 boys and 295 girls) aged 6 to 11 years 33tudes toward students with disabilities, rather than each parent’s attitude, contributed to a better understanding of the child’s attitudes toward students with disabilities. The theoretical and practical implications of these results are discussed.

## 1. Introduction

There is increasing international attention to school and social inclusion of children with disabilities, as it has been associated with their health-related quality of life [1]. The new social paradigm of inclusion underlines the contextual characteristics and variables that, when interacting with the personal health condition, can determine different levels of social and school participation and inclusion of children with disabilities in their life contexts [2]. Inclusion, in this perspective, focuses on the context; it highlights the obligation of living environments to enable everybody to actively participate and approach satisfying living within the given environment. For example, school inclusion of children with disabilities will significantly be determined by the school environment’s attitudes toward inclusion. The current study explored the possibility that school students’ parents are indirectly involved in shaping the attitudes toward the inclusion of a child with disability.

### 1.1. Environmental Determinants of School Inclusion

The existing literature mentions numerous factors that shape the process of school inclusion of students with disabilities, such as class size, teachers’ attitudes, and classmates’ attitudes [3]. The latter, in particular, has been considered one of the major barriers to inclusive education, as it negatively influences students with disabilities’ participation in their school contexts [4], up to dropping out of school [3]. Moreover, students’ negative attitudes toward their peers with disabilities might have crucial effects on students with disabilities that might possibly turn into problematic behavior, as a failure in school inclusion can sometimes lead to negative long-term consequences, such as depression, anxiety, loneliness, and other mental health problems [5]. Instead, positive attitudes could facilitate children with disabilities’ school inclusion, social acceptance, and health-related quality of life [6].

In an effort to approach a comprehensive understanding of the determinants of students’ attitudes toward disabilities, empirical studies were conducted while referring to the Theory of Planned Behavior [7]. These studies explored the possible role of personal (e.g., gender, age, experience with peers with disabilities) and contextual (e.g., parents’ or teachers’ attitudes toward people with disabilities) factors in shaping children’s attitudes toward peers with disabilities. A survey of the existing studies on school inclusion (see, for example, a meta-analysis by [8] and an international review by [3]) indicate that relatively much attention has been devoted to exploring the role of personal factors. These studies demonstrated that, for example, girls hold more positive attitudes than boys [9], and younger children hold more accepting attitudes toward their peers with disabilities than older children [10]. Characteristics of students with disabilities (i.e., type of disability) have also been extensively studied [3,8], showing that children hold especially negative attitudes toward students with behavioral problems (BP), more than toward students with sensory disability (SD) or intellectual disability (ID), due to the students with BP’s difficulties in managing proper social relationships [3]. As for contextual factors, in recent years, the number of studies on teachers’ attitudes toward students with disabilities and school inclusion has become quite large [11,12,13]. These studies highlight several factors that contribute to the teacher’s attitude toward students with disabilities, such as the teacher’s level of self-efficacy [14] and past training in special education [15].

However, only a few studies, mostly dated, have attempted to examine the relationships between parents’ attitudes and their child’s attitudes toward students with disabilities [16]; moreover, the studies that addressed this topic yielded inconsistent results. Thus, the current study aimed to contribute to a better understanding of the connection between parents’ attitudes and their child’s attitudes toward students with disabilities. In pursuit of this goal, the connection between the child’s and parents’ attitudes has been checked twice: (a) between the child’s attitude and each parent’s individual (i.e., separated) attitudes and (b) between the child’s attitude and the two parents’ combined attitude, as calculated by the level of similarity between the two parents’ attitudes. The study was conducted among a sample of Italian families with typically developing children who were questioned about their attitudes toward their (child’s) learning with peers with various disabilities.

### 1.2. Parents’ Impact on Their Child’s Attitudes toward Peers with Disabilities

Several theoretical approaches could have been suggested as the rationale to expect an impact of the father’s and mother’s attitudes upon their child’s attitude toward inclusion. Among these theories are Allport’s [17] Theory on Social Contact, Feinman’s [18] Theory of Social Referencing see [19], and Developmental Intergroup Theory [20]. These theories can be divided into two general groups: developmental perspective theories that predict that parent–child attitudes converge as the child becomes an adult, and socialization theories that suggest that the parent–child attitudes diverge over time [21]. Yet, all of these theories highlight that parents’ attitudes shape the child’s attitudes either in one direction or the other. Referring to attitudes toward inclusion, the most prominent and common basis for the general notion that parents play a significant role in establishing the child’s attitudes lies in traditional conceptions of children’s socialization, i.e., the idea that the father and the mother are the principal agents of socialization for their children’s social attitudes [22] and play a significant role in shaping their children’s attitudes toward peers with disabilities [10]. The parents deliver their attitudes [23] both directly, through explicit teaching, modeling, or discussions about relationships with—or attitudes toward—other people, or indirectly by providing the child with opportunities to interact with peers with disabilities and vulnerabilities. In addition, whether consciously or not, fathers and mothers reveal their values and beliefs about other people in their daily interactions with their children and others, as well as in their mutual interactions in front of their children.

However, not only is the number of studies that empirically tested the relationships between fathers’ and mothers’ attitudes and their child’s attitudes toward students with disabilities small, but those that did address this topic found mixed results. Where some studies [3,24] found that children of parents with positive attitudes toward inclusion were more welcoming toward peers with disabilities, other studies failed to repeat this finding [10,25,26]. Hence, various explanations have been suggested to outline under what terms parental impact could exist. For example, [27] argued that the child’s attitudes are connected to the mother’s attitudes only; [19] suggested that the child’s self-esteem may mediate parental impact on the child’s attitudes toward inclusion; [28] advocated that parental impact is mediated by the child’s general attitude toward one’s peers; finally, [10,27] claimed that parental impact on the child’s attitudes gained support for adolescents but not for younger children.

Notably, the validity of this age-related claim deserves a closer look. This is due to two reasons: (a) A review of the existing literature indicates that most of the studies on the impact of fathers’ and mothers’ attitudes on their children were conducted among young children, mostly ages 4–8 [10,19,26,27]. (b) The literature on the children–parents relationship suggests that a parent–child similarity in attitudes could be expected because young children tend to imitate their parents’ attitudes [22,29]. However, the literature on older children (i.e., late childhood and adolescence) tends to highlight the role of peers over, or at least to the same amount as, the parents’ role in shaping adolescents’ attitudes [28,30,31]. Moreover, a moderate level of children–parent disagreement is normative [32] and, in several terms, is even considered as a positive development outcome [33]. Hence, it can be assumed that, up to a certain level, the older the children get, the more discrepancies between their own and their fathers’ and mothers’ opinions are expected, especially those related to their social life [34]. Accordingly, the connection between the child’s and fathers’ and mothers’ inclusive attitudes is expected to be lessened rather than becoming more evident, as proposed above.

Yet, such an age-dependent differentiation assumes that the parent’s attitude is coherent and unified; what will happen once there is a discrepancy among the parents’ attitudes?

### 1.3. Inter-Parental Similarity and Their Child’s Inclusive Attitudes

When considering parents as agents of socialization of their children’s attitudes, [35] and, more recently, [36] suggested that it is also important to consider the inter-parental similarity variable; i.e., the level of agreement between the parents’ attitudes. Referring to inclusive education, though no studies have yet considered the role of parental similarity in shaping their child’s attitudes toward peers with disabilities, detection of the literature on children–parent relationships shows that recent studies are increasingly focused on exploring this topic from a family systems perspective [37] Accordingly, the dyadic interaction between parents is considered more than the sum of its parts and serves as a mechanism by which parents influence children’s development and attitudes [38].

Such a perspective investigates the family as a system composed of several subsystems; one of them is the level of coherence between the mother’s and the father’s attitudes. Namely, in general, or over a specific topic, the similarity between the two parents’ opinions and/or behaviors delivers a message (see below) that is different from the one delivered once each parent holds a different opinion [39]. Importantly, the similarity between the two parents’ messages is not necessarily always better or more accepted by their child; rather, in the eyes of the child, a coherent parental attitude would be perceived as more demanding or powerful.

Several studies support the notion that parental similarity (i.e., agreement) might have a singular impact on the child versus a lack of similarity. For example, a coherent restrictive parental discipline would be associated with a child’s greater aggression [40]; less parental similarity would be associated with a child’s lower adjustment [41]. However, parental similarity is not always a positive situation, as, under certain circumstances or specific family dynamics, parental disagreement may sometimes have positive effects on the children’s emotional and behavioral outcomes [42]. For example, [43] found that inter-parental conflict can lead to child’s emotional security when parental disintegration avoidance (i.e., the general tendency toward harmony) is high.

Regarding attitudes toward inclusion, the possible impact of inter-parental similarity on their child is unclear: On the one hand, parental similarity over either positive or negative attitudes toward inclusion is expected to encourage their child to adopt the same attitude. However, on the other hand, once expressed openly in the family, parental disagreement legitimizes the existence and acceptance of “the other” and “the different” person and opinions. This, in turn, may elicit more positive attitudes toward social inclusion. An example of this possibility comes from the literature on emotional socialization [44]; i.e., parental openness to child’s expressions of various emotions are positively related to the child’s development and ways of coping, while consistency in the parents’ negative reactions to the expression of certain emotions would lead to the child’s more “closeness”, up to psychopathology [45].

In her Emotion-Socialization Comprehensive Model, [44] highlights the direct and indirect effects of parents’ emotion-related socialization behaviors (ERSBs) on children’s outcomes through children’s affective arousal and self-regulation process and acknowledges the importance of the “variability and consistency of parental behavior” (p. 655) as a significant moderator variable that shapes a child’s social behavior and social competence. Moreover, as [46] demonstrated, investigating the family from a dynamic and dyadic perspective, in which the family is explored as a complex system rather than a (simple) sum of its parts, would make a unique contribution to understanding child’s emotional and social development. Considering these notions, a general assumption could be made, according to which each parent’s inclusive attitude, as the level of similarity between the two parents’ inclusive attitudes, shape their child’s inclusive attitudes. In light of the lack of studies on this topic and especially the possible variance in parent’s attitudes—i.e., as both parents’ attitudes could be either positive, negative, or mixed—currently, a two-sided assumption can be suggested, according to which a similarity or lack of similarity between the parents’ attitudes toward inclusion would play a moderating role in shaping the child’s attitudes toward inclusion.

### 1.4. The Present Study

There is still only a relatively limited body of knowledge regarding parental attitudes toward inclusion [16]. Therefore, this study aimed to examine the impact of both parents’ inclusive attitudes and the level of similarity between their attitudes on their child’s attitudes toward peers with disabilities. The type (i.e., direction) of similarity between the two parents’ attitudes is supposed to create a kind of family climate that encourages or discourages openness to others and, therefore, children’s attitudes toward students with disabilities. In addition, taking into account the evidence regarding the possible impact of the type of disability on others’ attitudes [3,8], attitudes toward three types of disabilities—i.e., attitudes toward a child with SD (hearing; SD), a child with ID (Down syndrome; ID), and a child with problems with aggressiveness and angry outbursts (BP) were measured and compared.

Specifically, the general hypothesis was that the degree of similarity in parents’ attitudes toward students with disabilities (and not the simple and direct effect of each parent’s attitudes) would moderate the relationship between fathers’ and mothers’ attitudes and their children’s attitudes toward other students with disabilities, be it SD, ID, or BP.

## 2. Materials and Methods

### 2.1. Participants

This study involved 598 Italian elementary school students aged 6 to 11 (M = 8.75, SD = 1.47) and their parents. Specifically, the sample was composed of 55 students (30 boys and 25 girls) attending 1st grade, 48 students (21 boys and 27 girls) attending 2nd grade, 154 students (66 boys and 88 girls) attending 3rd grade, 132 students (66 boys and 66 girls) attending 4th grade, and 209 students (120 boys and 89 girls) attending 5th grade.

The study participants were a non-probability sample of Italian parents. The pairs of parents and children were identified by contacting different Italian primary schools randomly selected within the Veneto region in northern Italy. First, a letter was sent to the school principals explaining the nature of the research project and requesting their participation. Once the primary school subscriptions were gathered, the school principals randomly selected the classes, and the students were invited to join the research project through parental consent and voluntary participation. Students could refuse participation in the research project without being segregated from their class (during the questionnaire administration). However, all the students of the selected classes participated in the research project. Parents were mailed a written invitation to participate in the research project and were informed that their answers would allow the drafting of a personalized report of their positive aspects, which was sent to them afterwards. Of those who accepted an invitation to participate, approximately 95% of the parents completed and sent the questionnaire back.

### 2.2. Measure

Children’s attitudes toward peers with disabilities were assessed by Nota and Soresi’s [47] questionnaire of students’ attitudes toward inclusion. The questionnaire presents descriptions of three hypothetical classmates with disabilities, the first characterized by an SD (hearing disability); the second by ID (Down’s syndrome); and the third by BP (problems with aggressiveness and angry outbursts). Out of Nota and Soresi’s [47] questionnaire, the respondents were requested to indicate their agreement with items that deal with behavioral intentions, as follows: “I would… (a) like to have her/him as my best friend; (b) like her/him to invite me to her/his house; (c) go to her/his house to play; (d) invite her/him to sleep at my house; (e) invite her/him to my birthday party; (f) keep her/him company during the breaktime at school; (g) have fun playing with her/him; (h) like her/him to be my neighbour; (i) talk to her/him; (j) like her/him to be my friend in the same way I like to be friends with other children”. Regarding each one of the items, the respondents indicated their attitudes on 5-point scale items (1 = strongly disagree, 5 = strongly agree) to examine the attitudes toward peers with disabilities. This procedure and measurement repeated itself for the three types of disabilities. The reliabilities (α) of this children’s scale for the three types of disabilities, were: Attitudes toward students with SD 0.85; Attitudes toward students with ID 0.88; and Attitudes toward students with BP 0.92.

Parents’ attitudes toward children with disabilities were measured by a questionnaire based on Ginevra et al.’s [48] study. Initially, this questionnaire assessed teachers’ attitudes toward students with disabilities. The instrument presents three different vignettes of a hypothetical child’s classmates with the following disabilities: the first vignette describes a child with an SD; the second describes a child with ID; and the third describes a child with BP. In line with the above-mentioned selection of behavioral items for the children’s scale, out of the teachers’ questionnaire, the parents were requested to indicate their evaluations of the following items: “(a) Her/his school performance would be..; (b) Her/his attention to detail would be…; (c) The chance for her/him to complete a classwork alone would be…; (d) The chance for her/him to adequately perform school activities would be…; (e) The chance for her/him to be admired by others would be…; (f) She/he would be perceived as “a resource” for the school. Using the instrument in a pilot study with 320 parents, a mono-factorial solution was found. In the present study, the reliabilities for the three types of disabilities (α) were as follows: fathers’ attitudes toward students with SD 0.74, ID 0.74, BP 0.76; mothers’ attitudes toward students with SD 0.74, ID 0.79, and BP 0.72.

### 2.3. Design

The study used a dyadic design, looking at pairs of parents and children.

### 2.4. Procedure

Upon receiving written consent from the parents and the parent-completed questionnaire, the children’s questionnaire was administered in the schools, with the students sitting in their home classes with their classmates. The questionnaire administration to the students was conducted by a professional psychologist who is experienced in group questionnaire administration procedures so that the children’s understanding of the items could be verified, and any questions could be answered individually. For children that were unable to write or read the items, showed learning disabilities, or had difficulty reading or writing in the Italian language, the questionnaire was administered individually by the researcher using an interview form.

Ethics Statement. No IRB format is required. All procedures performed in this study were in accordance with the ethical standards of the Italian Association of Career Guidance (SIO) and Italian Association of Psychologist (AIP). Specifically, according to the ethical code of the Italian Association Psychology approved in 2015 and revised in 2022 the study was developed and implemented respecting all rules of conduct under the code of ethics: information and consent for participation in research (article 1); return of results (article 3) through a personalized report for each participant; respect of privacy and anonymity (article 4).

Informed consent: Informed consent was obtained from all individual participants included in the study.

### 2.5. Data Analysis

#### 2.5.1. Preliminary Analysis

All assumptions for statistical analyses (i.e., normality, homogeneity of variance, sphericity, multicollinearity, outliers, and independence of observations) were checked. Distributions of all variables were evaluated in terms of mean, Standard Deviation, median, range, skewness, and kurtosis. No data imputation took place because there were no missing data. Moreover, to find out about collinearity, VIF estimation was calculated. Data showed a VIF score less than 5, which means that the model has no collinearity.

#### 2.5.2. Exploration of Simple Parental Effects

In order to explore the impact of parental similarity on a child’s attitudes, the following steps were performed: To check the alternative possibility to the study’s general hypothesis—i.e., that each parent has a simple and direct impact on the child’s attitudes—we computed Pearson correlations between the parents and their child’s attitudes. Following that, we performed three hierarchical regression analyses, each referring to one of the three domains of disability. The child’s personal data (gender and age) were entered in the hierarchical regression analyses in the first step. Following that, each parent’s attitude (separately) and, in addition, each parent’s attitude multiplied by itself were entered in the second step. Next, the parent’s multiple scores were included based on [49] recommendation to check possible quadratic relationships between the parents’ and the child’s attitudes. Finally, the multiplication of the father X mother attitudes was entered in the third step to check for possible interaction effects upon the child’s attitude. All steps were performed stepwise.

#### 2.5.3. Exploration of a Parental Similarity Effect

(1) Division into Sub-Groups: In light of the differences between the items that composed the parents and the children scales, and in pursuit of exploring the contribution of parental (lack of) opinion similarity, both the parents sample and children sample were divided, separately, into three groups—i.e., (a) High: the parents and/or the child express opinions that represent a relatively high level of attitude toward inclusion; (b) Average: the parents and/or the child hold an average positive attitude toward inclusion; and (c) Low: the parents and/or the child hold relatively low positive attitudes toward inclusion. Division of the parents and the children samples into these sub-groups was based on the whole (parents OR children) sample, [M +/− (½ SD)] (Notably, a division based on [M +/− (1 SD)] was impossible as it yielded several sub-groups with too small a number of children in each group/cell). These divisions yielded a 3 (parents groups) × 3 (children groups) matrix. Such a division was conducted concerning each of the three domains of disabilities—i.e., sensory, intellectual, and behavioral—separately.

(2) Level of Similarity: The rate of children who hold the same sub-group opinion as either their father or mother was then computed.

(3) Establishment of the Similarity Index: Based on division into sub-groups, a measure of parental similarity in opinions was established, with four nominal groups: Similarity-High: both parents hold strong positive attitudes toward inclusion; Similarity-Average: both parents hold medium attitudes; Similarity-Low: relatively speaking, both parents do not support inclusion; No-Similarity: the parents’ attitudes are not overlapping. The lack of similarity could be in all directions—i.e., the parents’ attitudes are either opposite or in the same direction but not to the same level of attitude.

(4) Exploration of the Similarity Index on Children’s Attitudes. To further explore whether the similarity between parents’ attitudes makes any difference in their child’s attitudes, the following steps were made: (a) three canonical linear discriminant functions were computed, each one for a different disability domain, to check the differences between the four parental groups (see above); (b) three two-way MANCOVAs were performed to check the direction of differences between the four parental groups. Specifically, to check what are the implications of parental similarity in attitudes, three two-way MANCOVAs were performed separately for each one of the three disability domains with the child’s gender and parental similarity group (see above) as the independent variables, the child’s age as a covariate, and the child’s responses to the relevant questionnaire items as the dependent variables.

#### 2.5.4. Exploration of the Moderating Effect

To check the study hypothesis regarding the moderating role of parents’ level of similarity in shaping the child’s attitudes toward inclusion, three SEMs analyses were performed to test the structural relations. In the final stage, we divided the whole sample into similarity index sub-groups and generation groups for each type of disability. Then, as our model is a typical structural model, we employed a multi-group analysis in SEM to test the moderation role of parental similarity in attitudes toward disability in the effects of mothers’ attitudes toward students with disabilities, fathers’ attitudes toward students with disabilities, and children’s gender and age on children’s attitudes toward peers with disabilities.

Multi-group function in MPLUS was used to conduct the group difference analysis. To test the direct associations between parents’ attitudes and children’s attitudes toward students with disabilities and the moderation effect of parental similarity, the model in Figure 1 was specified for the three types of disability (SD, ID, BP). In addition, two demographic variables, children’s gender and age, were included in this model.

As shown in Figure 1, to assess whether the relationships between the variables varied across the similarity index, a multi-group analysis was used. A multi-group analysis permits the model estimation simultaneously for the four groups. As a result, the parental similarity in the opinion groups (Similarity-Low group, Similarity-Average group, Similarity-High group, No-Similarity group) was well balanced and sufficiently large for structural modeling purposes in the different types of disability.

Across all of the analyses, we used four goodness-of-fit indices commonly employed in SEM: The Comparative Fit Index (CFI; values close to 0.95 or greater suggest a good fit), the Tucker–Lewis Index (TLI; values close to 0.95 or greater indicate a good fit), the Root Mean Square Error of Approximation (RMSEA; 0.06 or less denotes a good fit) with a 90% confidence interval (CI), and the Standardized Root Mean Square Residual (SRMR; 0.08 or less suggests an adequate fit).

## 3. Results

### 3.1. Exploration of Simple Parental Effects

Means, SDs, and correlations for the children’s and the parents’ scales are presented in Table 1. Generally speaking, the data presented in Table 1 indicated that the father’s and the mother’s attitudes toward students with disabilities are significantly related to each other. However, the children’s attitudes are not significantly correlated to those of the parents, except for a slight relationship between the child’s attitudes toward peers with SD and ID and the mother’s attitudes toward students with ID.

The results of the three hierarchical regression analyses to check whether parents’ attitudes contribute to explaining variance in children’s attitudes are presented in Table 2.

The results of the hierarchical regression analyses showed that the main predictors of a child’s attitudes were gender and age. The parents’ attitudes, both in their simple form or in their advanced quadric or interactive effects, had almost no marginal contribution. The only exception was the mother’s significant contribution to explaining the variance in the child’s attitudes toward students with ID.

### 3.2. Exploration of Parental Similarity Effects

As detailed above (see Section 2.5), the parents and the children samples were (separately) divided into three levels of attitudes (i.e., Low, Average, High) toward each one of the three disability domains. These divisions yielded the following ranges of sub-group means: for the fathers: SD: +4.90/average: 4.89–4.03/4.02; ID: +4.74/average: 4.73–3.89/3.88; BP: +4.43/average: 4.42–3.54/3.53; for the mothers: SD: +4.93/average: 4.92–4.09/4.07; ID: +4.87/average: 4.86–4.02/4.03; BP: +4.53/average: 4.52–3.62/3.63; and for the children: SD: +4.06/average: 4.05–3.36/3.37; ID: +4.12/average: 4.11–3.33/3.32; BP: +4.14/average: 4.13–2.99/2.98.

Based on these divisions, the rate of overlapping assignments between the children and each one of their parents was computed and is presented in Table 3.

The data presented in Table 3 shows that the moderate of child–parent overlapping assignments is around 70% but lower regarding the child–mother attitudes regarding the inclusion of a student with BP. Namely, despite the transition to a more loose definition of child–parent similarity (i.e., a change from precise data computation to a global, group-related evaluation of the level of similarity), a relatively large group of children (30%) hold different inclusive attitudes from their parents.

Discriminant analyses were performed to explore possible differences of the following four groups: (1) Similarity-Low: both parents agreed upon low support of inclusion (N = 97–126, in the three domains of disability); (2) Similarity-Average: the two parents agreed upon average support of inclusion (N = 120–157); (3) Similarity-High: both parents agreed upon high support of inclusion (N = 107–115); and (4) No-Similarity: the two parents disagreed between themselves about inclusion, at least concerning one of the three domains and at any level of disagreement (e.g., High vs. Average) (N = 211–253). The variables that were entered into the discriminant analyses were the nine items of the children’s scale (see Section 2.2).

Table 4 presents the structure matrix of the three functions received for each one of the disability domains. Alongside the eigenvalues, the variance rate is explained as well as the Wilk’s Lambda results. As presented in Table 4, the results of the analyses indicate that, though not a few of the canonical discriminant functions are satisfactory (>0.30), only one function is significant.

This function relates to the SD domain and is composed of the following variables (i.e., children’s questionnaire items: “I would talk to her/him” (0.816); “I would like her/him to be my neighbour” (0.336); “I would have fun playing with her/him” (0.337); “I would go to her/his house to play” (0.431); “like her/him to be my friend in the same way I like to be friends with other children” (0.449). However, more importantly, functions at the group centroid, as presented in Table 5, highlight the difference between the No-Similarity group and the other three groups of similarity; i.e., the first discriminant functions on both the SD domain and the ID domain, and to some degree also the BP domain, applied most strongly to the prediction of belonging to the No-Similarity group of parents versus the other three groups.

In pursuit of checking possible age and gender differences, MANCOVAs were performed on the children’s attitudes. Results of the three two-way MANCOVAs were as follows: for the SD domain, Wilks’ Lambdas were significant for age, *F*_age_ = 2.02, ƞ^2^_p_ = 0.034, *p* = 0.03; gender, *F*_gender_ = 3.63, ƞ^2^_p_ = 0.059, *p* = 0.03; and the similarity groups, *F*_groups_ = 1.66, ƞ^2^_p_ = 0.028, *p* = 0.014. The gender X group was not significant. For the ID domain, Wilks’ Lambdas were significant for age, *F*_age_ = 2.60, ƞ^2^_p_ = 0.063, *p* = 0.001, and gender, *F*_gender_ = 2.07, ƞ^2^_p_ = 0.146, *p* = 0.025. The Wilks’ Lambdas for both the similarity groups and gender X group were not significant. For the BP domain, Wilks’ Lambdas were significant for age, *F*_age_ = 3.55, ƞ^2^_p_ = 0.034, *p* < 0.03, and gender, *F*_gender_ = 8.97, ƞ^2^_p_ = 0.059, *p* < 0.03. The Wilks’ Lambdas for both the similarity groups and gender X group were not significant.

Table 6 presents the means and SDs and results of the comparisons between the parents’ similarity groups for those children’s items that yielded significant differences in the MANCOVA analysis. Sceffe’s post hoc multiple comparisons (*p* < 0.05) yielded significant differences between the Similarity-Low group and the No-Similarity group for the following items, “I would have fun playing with her/him” and “I would talk to her/him”, in the SD domain. However, in light of the exploratory nature of the current study, we also computed LSD post hoc multiple comparisons and found that the significant difference between the Similarity-Low and the No-Similarity groups is comprehensive and repeatedly found across many of the children scale’s items. Notably, across the three disability domains, the difference between the Similarity-Low and the No-Similarity groups was found for the following items: “I would go to her/his house to play”, “I would like her/him to be my neighbour”, and “I would like her/him to be my friend in the same way I like to be friends with other children”.

### 3.3. Exploration of Moderating Effects

Moderation effects were tested through multi-group analyses of structural equation modeling analyses (SEM). These analyses yielded several significant moderation effects, as follows:

(a) Attitudes toward students with SD (Figure 2): (1) In the No-Similarity and Similarity-Low groups, a link was found between children’s gender (β_no-similarity group_ = 0.02, *p* = 0.003; β_similarity-low group_ = 0.24, *p* = 0.005) and children’s attitudes toward peers with SD. (2) In the Similarity-Average group: children’s gender (β = 0.02, *p* = 0.005) and father’s attitudes toward children with SD (β = 0.65, *p* = 0.008) had a significant impact on children’s attitudes toward peers with SD. (3) In the Similarity-High group, there was a significant connection between the children’s age and their attitudes toward peers with SD (β = −0.18, *p* = 0.005).

(b) Attitudes toward students with ID (Figure 3): in the No-Similarity and Similarity-Low groups, but not the Similarity-Average and Similarity-High groups, a link between age (β_no-similarity group_ = −0.15, *p* = 0.003; β_similarity-low group_ = −0.12, *p* = 0.021) and children’s attitudes toward peers with ID was found.

(c) Attitudes toward students with BP (Figure 4): (1) in the No-Similarity, Similarity-Average, and Similarity-High groups, a link was found between children’s age (β_no-similarity group_ = −0.18, *p* = 0.022; β_similarity-average group_ = −0.17, *p* = 0.016; β_similarity-high group_ = −0.19, *p* = 0.038), gender (β_no-similarity group_ = −0.02, *p* = 0.001; β_similarity-average group_ = 0.02, *p* = 0.001; β_similarity-high group_ = 0.03, *p* = 0.001), and their attitudes toward peers with BP. (2) In the Similarity-Low group, significant relationships between the children’s gender (β = 0.03, *p* = 0.001) and age (β = −0.22, *p* = 0.016), the fathers’ attitudes toward children with BP (β = 0.27, *p* = 0.002), and the mothers’ attitudes toward children with BP (β = 0.32, *p* = 0.010) were found.

## 4. Discussion

This study aimed to test the impact of fathers’ and mothers’ attitudes and the level of similarity between their attitudes on their children’s attitudes toward peers with Sd, ID, and BP. The study findings suggest preliminary support to the study’s general hypothesis, according to which the inter-parental similarity of attitudes toward students with disabilities, rather than each parent’s individual attitude, contributes to a better understanding of the child’s own attitudes toward students with disabilities. Below is an elaboration of the study findings and their meaning.

In order to approach the study goals, data were collected among a large sample of children and their parents, followed by various statistical analyses. Firstly, to test the simple and direct impact of fathers’ and mothers’ attitudes toward students with disabilities on the child’s attitudes (measured in terms of behavioral intention), three hierarchical regression analyses (each one refers to one of the three domains of disability) were computed. It was found that each parent’s individual attitudes had almost no marginal contribution to explaining a child’s attitudes toward peers with disabilities. Rather, the main predictors of a child’s attitudes were gender and age. Specifically, girls reported more inclusive attitudes toward peers with disabilities than boys. This result replicates the findings of previous research [9,32]. Girls’ more favorable attitudes toward inclusion could be explained as an adhesion to common gender stereotypes, as girls are expected [50] to show more concern and understanding for others, especially toward individuals with difficulties, and social and emotional sensitivity compared to boys. This is true especially for Italian girls, as they are prematurely exposed, e.g., through textbooks and social role models, to those gender stereotypes.

Regarding age, the results of this study showed that children’s negative attitudes toward peers with disabilities increase with age. This result is also reported in previous research [9] and could be explained by considering children’s cognitive development. Namely, the cognitive components, which are necessary for the prejudice to emerge as a mindset and attitudes toward individuals of the outgroup (i.e., individuals with disabilities) [51], are gradually acquired along the child’s development and could lead to the emergence of solid prejudice around ten years of age [52].

Secondly, to explore the parental similarity effect on children’s attitudes, three canonical linear discriminant functions and three MANCOVAs were performed separately for each one of the three disability domains. The parental similarity effect on children’s attitudes was significant in a small way only on SD domain, showing that children have a greater intention to have fun playing and talk with hypothetical classmates with SD when their parents disagreed between themselves about inclusion than peers whose parents agreed upon low support of inclusion. However, notably, LSD post hoc multiple comparisons found a significant difference between the Similarity-Low and the No-Similarity groups, across the three disability domains, in many of the children scale’s items. In line with the preliminary theoretical reliance on the Emotion-Socialization Comprehensive Model [44], it is plausible to suggest that when parents disagree between themselves on matters related to the inclusion of students with disabilities, their child may become more open-minded toward heterogeneity in general and specifically in the school context, which can foster more positive attitudes toward classmates with disabilities.

Thirdly, multi-group modeling analyses were performed to further explore the moderating role of parents’ level of similarity in shaping the child’s attitudes toward inclusion. These analyses showed that parents’ level of similarity moderates the relationship between the father’s simple and direct attitudes and the child’s attitudes toward peers with SD and BP once both parents held either an average (i.e., Similarity-Average group) or low (i.e., Similarity-Low group) levels of support in inclusion. The simple and direct effect of mothers’ attitudes on their children’s attitudes was moderated by parents’ level of similarity only in the BP domain and once the parents agreed upon low support of inclusion (Similarity-Low group).

Overall, these results further show that the simple and direct effect of mothers’ or fathers’ attitudes positively affect children’s attitudes when there is low to medium similarity between the parents, thus supporting the assumption that, beyond the (separated) fathers’ and mothers’ attitudes toward inclusion, it is important to also consider the parents as a couple, a variable that has an additional significance beyond the one that could be attributed to each one of them independently. As mentioned above, a possible explanation for these results is that the low or average similarity between parents can foster in the children a more openness toward “the other” and, hence, to more positive attitudes toward peers with SD or BP. It is also possible that the low constructive similarity between parents may have led their children to a better comprehension of other people’s emotions and positive social behavior and, therefore, more positive attitudes toward peers with these disabilities [42]

However, the results for the case of a peer/student who has an ID are different. A moderating effect of the parents’ level of similarity on shaping the child’s attitudes toward peers with ID was not found. This result is in line with several other studies that failed to find a relationship between parents and their children’s attitudes toward peers with ID [10,25,26]. A possible explanation to the differential attitude toward the topic of ID relates to the “presence” of this disability in common family discourses; i.e., topics related to ID, and in particular Down syndrome, are hardly and poorly addressed in the family context [53] and, therefore, the ideas and attitudes that shape children’s attitudes toward others who have this disability emerge on different grounds rather than the parents’ individual attitudes and the combined (e.g., similarity) attitudes of their parents.

Several limitations of this study were identified. Firstly, only self-reported data were used to analyze attitudes. Further studies could use both direct and indirect measures to examine attitudes. Secondly, non-equivalent measures were used to assess children’s and their parents’ attitudes toward disabilities. Although we have tried to check as much as possible what this source of error was, in future studies, a specific measure for assessing children’s and their parents’ attitudes could be developed. Thirdly, in this study, we focused only on children’s and parents’ attitudes toward peers with three types of disabilities; future studies could expand the scope of interest and examine other disabilities and vulnerabilities, such as immigration and low socioeconomic status associated with disability. Fourthly, the impact of parents and children on the health-related quality of life of students with disabilities was not examined and should be assessed in further studies.

However, in spite of these limitations, the study findings have both theoretical and practical value. In terms of theoretical implications, the study findings foster the establishment of a more efficient model of parental impact on a child’s inclusive attitudes and behavior. This is due to two reasons: Firstly, the study findings demonstrate that explorations of each parent’s individual impact on the development of inclusive attitudes are marginal and seem to be of limited future utility. Secondly, the present study results highlight the importance of the dyadic interaction between parents that plays a moderating role in shaping their child’s attitudes toward inclusion. Hence, the reliance of existing models of the family as a system, rather than an additive impact of individuals, seems to have the potential for significantly promoting the establishment of an efficient theoretical model of the development of children’s inclusive behavior.

In terms of practical implications, the study stimulates professionals to consider appropriate strategies to promote children’s and parents’ attitudes toward disabilities to include and support people with disabilities and promote their health-related quality of life. Firstly, it suggests taking preliminarily into consideration children’s gender and age along with the type of disability of peers with disabilities in the implementation of preventive interventions aimed at promoting inclusive education and positive attitudes toward peers with disabilities. Secondly, it also suggests that in addition to working with children, there is a need to address the family context—i.e., how the issues related to disability, diversity, and inclusion are discussed in the familial context and the degree of freedom that parents leave their child to make positive interpersonal experiences toward disabilities. Interventions aimed at fostering children’s attitudes toward peers with disabilities should involve parents, as this may lead to greater effectiveness [26]. In this regard, particular attention should be given to parenting couples, as the dyadic interaction between parents serves as a mechanism by which parents influence the development of attitudes in children. Intervention activities should not pay homage to parents’ attitudes but rather encourage parents to have a constructive discussion/conflict in a way that will provoke more positive emotional and behavioral responses in children [42].

## 5. Conclusions

Classmates’ attitudes are being considered one of the major barriers to inclusive education, as it negatively influences students with disabilities’ participation in their school contexts. The present study showed that the inter-parental similarity of attitudes toward students with disabilities, rather than each parent’s attitude, contributed to a better understanding of the child’s attitudes toward students with disabilities. Positive attitudes could facilitate children with disabilities school inclusion, social acceptance, and health-related quality of life.

## Figures and Tables

**Figure 1 ijerph-19-16387-f001:**
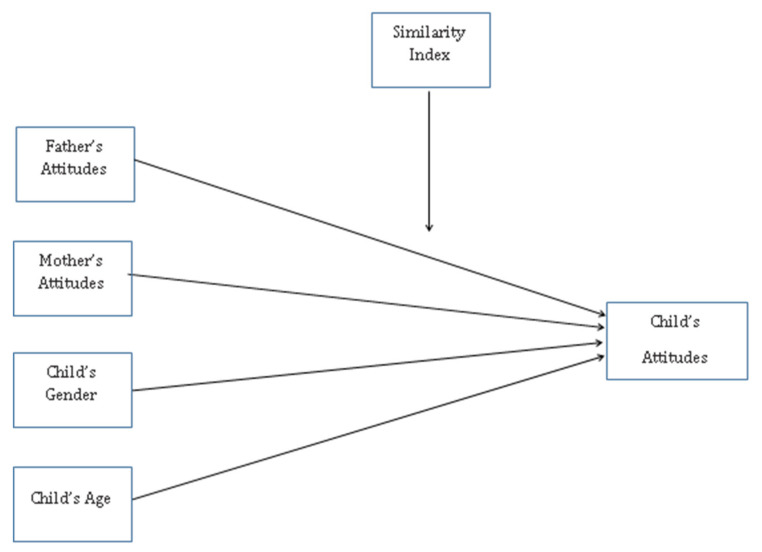
Predicted Model.

**Figure 2 ijerph-19-16387-f002:**
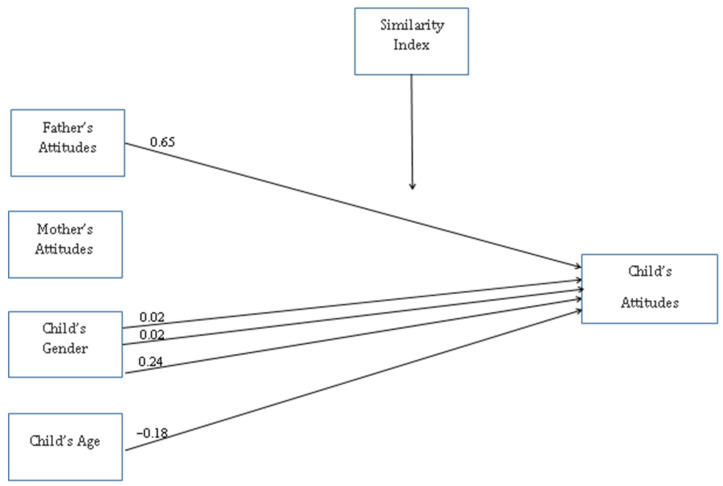
Attitudes toward children with SD.

**Figure 3 ijerph-19-16387-f003:**
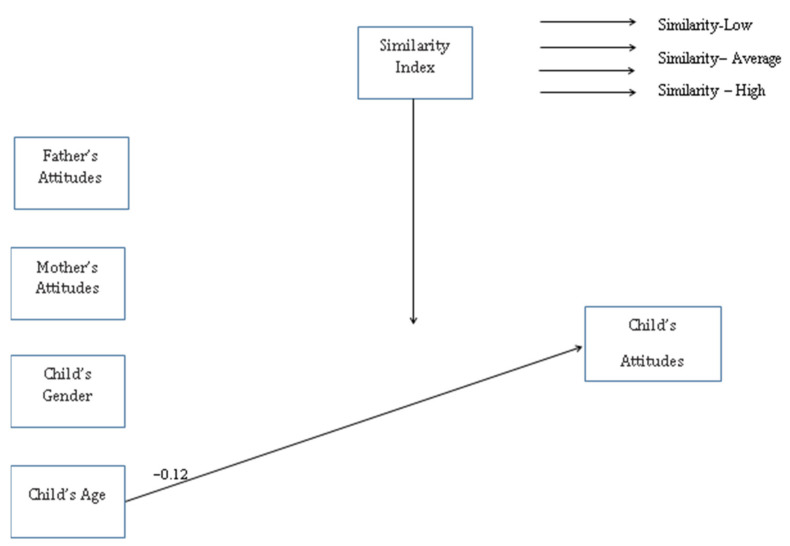
Attitudes toward children with ID.

**Figure 4 ijerph-19-16387-f004:**
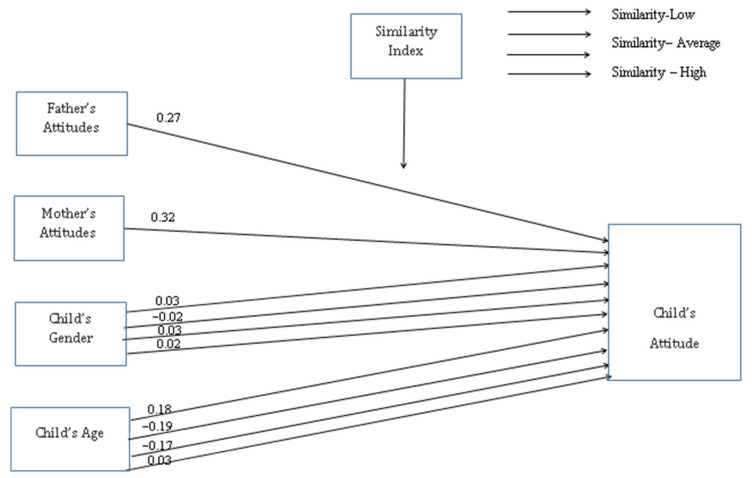
Attitudes toward students with BP.

**Table 1 ijerph-19-16387-t001:** Pearson correlations among dimensions.

				Father			Mother			Child	
		M	SD	SD	ID	BP	SD	ID	BP	SD	ID
Father											
	SD	4.46	0.882								
	ID	4.31	0.872	0.691 **							
	BP	3.98	0.895	0.433 **	0.443 **						
Mother											
	SD	4.50	0.866	0.579 **	0.422 **	0.293 **					
	ID	4.45	0.852	0.408 **	0.544 **	0.292 **	0.681 **				
	BP	4.08	0.896	0.270 **	0.281 **	0.553 **	0.417 **	0.450 **			
Child											
	SD	3.73	0.722	0.043	0.049	0.049	0.040	0.091 *	0.053		
	ID	3.72	0.795	0.066	0.065	0.042	0.051	0.110 **	0.048	0.700 **	
	BP	3.46	0.955	0.031	0.000	0.018	−0.013	0.018	0.016	0.513 **	0.431 **

Note: * *p* < 0.01, ** *p* < 0.001.

**Table 2 ijerph-19-16387-t002:** Results of Hierarchical Regression Analyses showing the amount of variance in a child’s attitudes toward students with disabilities accounted for by the simple and combined effects of parents’ attitudes.

		Model 1			Model 2		
		B	SD E	β	B	SD E	β
**Predicted Variable: Attitudes toward a Student with SD**							
Step 1	Gender	0.317	0.058	0.220 ***	0.302	0.057	0.302 ***
	Class	−0.089	0.022	−0.158	−0.089	0.022	−0.158 ***
Step 2	Father’s attitude				0.053	0.287	0.179
	Mother’s attitude				0.061	0.298	−0.250
	(Father’s attitude)^2^				−0.052	0.549	−0.760
	(Mother’s attitude)^2^				0.064	0.536	0.145
Step 3	Father’s attitude X Mother’s attitude				−0.056	0.118	−1.712
	(Father’s attitude X Mother’s attitude)^2^				0.051	0.001	0.401
	Adjusted *R*^2^		0.048			0.070	
	*F*		30.28 ***			23.54 ***	
	Δ*R*^2^					0.025	
	Δ*F* statistics					16.05 ***	
**Predicted Variable: Attitudes toward a Student with ID**							
Step 1	Class	−0.106	0.025	−0.171 ***	−0.105	0.025	−0.169 ***
	Gender				0.079	0.065	0.079
Step 2	Father’s attitude				0.033	0.529	0.788
	Mother’s attitude				0.099	0.037	0.106 **
	(Father’s attitude)^2^				−0.025	0.045	−0.483
	(Mother’s attitude)^2^				0.010	0.048	0.008
Step 3	Father’s attitude X Mother’s attitude				−0.042	0.113	−0.892
	(Father’s attitude X Mother’s attitude)^2^				0.025	0.001	0.440
	Adjusted *R*^2^		0.029			0.037	
	*F*		18.01 ***			11.52 ***	
	Δ*R*^2^					0.011	
	Δ*F statistics*					6.98 **	
**Predicted Variable: Attitudes toward a Student with BP**							
Step 1	Gender	0.673	0.073	0.352	0.651	0.072	0.341
	Age				−0.122	0.028	−0.164
Step 2	Father’s attitude				0.024	0.519	1.061
	Mother’s attitude				−0.026	0.531	−0.011
	(Father’s attitude)^2^				−0.009	0.051	−0.762
	(Mother’s attitude)^2^				0.027	0.053	0.361
Step 3	Father’s attitude X Mother’s attitude				−0.019	0.114	−0.886
	(Father’s attitude X Mother’s attitude)^2^				0.006	0.002	0.362
	Adjusted *R*^2^		0.123			0.148	
	*F*		84.548 ***			52.903 ***	
	Δ*R*^2^					0.027	
	Δ*F* statistics					18.742 ***	

Note: ** *p* < 0.01, *** *p* < 0.001.

**Table 3 ijerph-19-16387-t003:** Level of similarity between child’s and parents’ attitudes regarding each domain of disability.

			Father			Mother		
			Low	Average	High	Low	Average	High
Child’s attitudes		Low	**14**	72	14	**13**	71	16
	SD	Average	17	**69**	14	15	**70**	15
		High	17	68	**15**	16	69	**16**
		Low	**14**	78	8	**20**	68	11
	ID	Average	13	**71**	16	16	**68**	17
		High	11	69	**19**	13	60	**26**
		Low	**16**	64	20	**28**	36	36
	BP	Average	14	**72**	14	29	**41**	29
		High	10	67	**22**	33	32	**35**

Note: Bolded numbers indicate similarity between the child’s and either the father’s or the mother’s group-level of attitudes toward inclusion.

**Table 4 ijerph-19-16387-t004:** Structure matrix emerging from the discriminant function analysis for the three disability domains.

	SD			ID			BP		
	1	2	3	1	2	3	1	2	3
I would talk to her/him	**0.816 ***	−0.127	0.019	**0.622 ***	0.086	0.014	0.245	**0.724 ***	−0.147
I would like her/him to be my neighbour	**0.336 ***	0.329	0.253	**0.423 ***	−0.145	0.202	−0.039	0.214	**0.389 ***
I would have fun playing with her/him	0.337	**0.619 ***	0.343	0.200	0.117	**0.534 ***	**0.154 ***	0.029	−0.034
I would like her/him to invite me to her/his house	−0.011	**0.373 ***	0.230	0.139	−0.344	**0.524 ***	**386 ***	−0.080	−0.018
I would invite her/him to sleep at my house	−0.023	**0.295 ***	−0.121	−0.210	0.465	**0.532 ***	−0.143	**−0.188 ***	0.077
I would invite her/him to my birthday party	−0.054	−0.093	**0.621 ***	−0.144	−0.282	**0.525 ***	**0.244 ***	0.089	0.183
I would go to her/his house to play	0.431	0.097	**0.519 ***	0.332	−0.009	**0.764 ***	**511 ***	−0.046	0.168
I would like her/him to be my friend in the same way I like to be friends with other children	0.449	0.292	**0.450 ***	**0.613 ***	−0.090	0.462	**547 ***	0.243	0.015
I would keep her/his company during the breaktime at school	0.268	0.121	**0.368 ***	0.059	0.029	**0.494 ***	0.338	0.235	**0.533 ***
I would like to have her/him as my best friend	−0.026	0.068	**0.317 ***	−0.097	0.069	**0.386 ***	**0.141 ***	−0.109	0.096
Eigenvalue	0.49	0.22	0.16	0.029	0.022	0.016	0.022	0.015	0.010
% of variance explained	56.3	25.5	18.1	42.6	33.2	24.2	46.9	30.9	22.2
Wilk’s Lambda (df = 30)	0.917 **	0.963	0.984	0.935	0.962	0.984	0.954	0.975	0.990

Note: * *p* < 0.01, ** *p* < 0.001.

**Table 5 ijerph-19-16387-t005:** Functions at group centroids for the three disabilities domains.

	SD			ID			BP		
Parental Similarity on Inclusion	1	2	3	1	2	3	1	2	3
Similarity-Low	−0.068	−0.261	−0.097	−0.299	0.142	−0.082	−0.321	0.081	0.014
Similarity-Average	−0.099	0.176	−0.152	−0.009	−0.264	−0.117	0.061	0.021	−0.164
Similarity-High	−0.313	0.024	0.190	−0.065	−0.066	0.250	−0.025	−0.253	0.035
No-Similarity	0.278	0.020	0.062	0.164	0.094	−0.023	0.102	0.067	0.087

**Table 6 ijerph-19-16387-t006:** Means, SDs, and F values for items discriminating between the types of parental similarity.

Parental Level of Similarity		Similarity-Low	Similarity-Average	Similarity-High	No-Similarity	Total	*F_groups_*	ƞ^2^_p_
SD								
I would go to her/his house to play	M	3.60	3.61	3.71	3.65	3.64	3.02 *	0.015
	(SD)	(1.01)	(1.00)	(0.97)	(0.91)	(0.96)		
I would have fun playing with her/him	M	3.45	3.70	3.65	3.80	3.67	3.61 **	0.018
	(SD)	(1.10)	(0.97)	(0.96)	(0.99)	(1.01)		
I would talk to her/him	M	3.77	3.68	3.50	4.06	3.80	6.04 ***	0.030
	(SD)	(1.15)	(1.21)	(1.28)	(1.02)	(1.16)		
I would like her/him to be my friend in the same way I like to be friends with other children	M	3.48	3.59	3.61	3.87	3.67	3.01 *	0.015
	(SD)	(1.32)	(1.33)	(1.31)	(1.07)	(1.24)		
ID								
I would go to her/his house to play	M	3.73	3.83	3.96	3.45	3.68	2.96 *	0.015
	(SD)	(1.18)	(1.13)	(1.04)	(1.21)	(1.19)		
I would like her/him to be my friend in the same way I like to be friends with other children	M	3.38	3.63	3.78	3.78	3.68	2.81 **	0.014
	(SD)	(1.41)	(1.35)	(1.28)	(1.15)	(1.27)		
BP								
I would go to her/his house to play	M	3.26	3.46	3.47	3.54	3.46	4.53 **	0.23
	(SD)	(1.34)	(1.29)	(1.18)	(1.16)	(1.23)		

Note: * *p* < 0.05, ** *p* < 0.01, *** *p* < 0.001.

## Data Availability

The data presented in this study are available on request from the corresponding author. Please contact mariacristina.ginevra@unipd.it.

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
