# Peer review of "Inclusive Socialization? The Relationships between Parents’ and Their Child’s Attitudes toward Students with Disabilities"

_ijerph, 2022, doi:10.3390/ijerph192416387_

Round 1

Reviewer 1 Report

The article presents a study that pursues valid objectives, also the results obtained are interesting and can contribute to the improvement of the processes of inclusion of people with disabilities. The methodological approach is appropriate, although it would have been more accurate to have a larger sample of mothers and fathers. This situation unbalances the two groups that make up the sample. In addition, given such a small number of fathers and mothers, it is questionable that a questionnaire be used to measure opinions. Despite these points, it is considered very difficult to improve them and, therefore, the article can be published in the current format.

Author Response

Thank you for your positive feedback

Reviewer 2 Report

The author mentioned that "No IRB format is required from XXX because there are no experiments on individual participants and the study has a theoretical and preventive implication."

I have doubt whether it aligns with the journal's policy as this kind of questionnaire study normally requires ethical approval. It also involves research participants of minors (i.e., 6-11 years old).

I have also attached the following instructions from the journal's website: "For non-interventional studies (e.g. surveys, questionnaires, social media research), all participants must be fully informed if the anonymity is assured, why the research is being conducted, how their data will be used and if there are any risks associated. As with all research involving humans, ethical approval from an appropriate ethics committee must be obtained prior to conducting the study. If ethical approval is not required, authors must either provide an exemption from the ethics committee or are encouraged to cite the local or national legislation that indicates ethics approval is not required for this type of study. Where a study has been granted exemption, the name of the ethics committee which provided this should be stated in Section ‘Institutional Review Board Statement’ with a full explanation regarding why ethical approval was not required."

Author Response

Thanks for your revision, we provide national legislation that indicates ethics approval is not required for this type of study.

Reviewer 3 Report

The topic of the manuscript is within the scope of the Journal and could be valuable to the scientific audience. The quality of the research design is acceptable.

The research hypothesis is formulated.

 TITLE

The title of the article is accurate.

ABSTRACT

Abstract reflects the work done and the conclusions drawn.

INTRODUCTION

Clarify how the present research makes a novel contribution to the literature.

METHOD

Some clarifications are however needed.

Some assumptions are required for ANCOVA (normality, homogeneity of variance, multicollinearity, sphericity). In addition, ANCOVA requires the following additional assumptions: For each level of the independent variable, there is a linear relationship between the dependent variable and the covariate. Why is it not justified whether conditions for using ANCOVA were met (for instance, normality, multicollinearity, sphericity, homogeneity of variance)?

If ANCOVA were performed partial Eta-squared must be calculated but not Eta-squared (ηp2).

RESULTS

Some clarifications are however needed.

Effect sizes for F-statistics must be expressed as partial Eta-squared (ηp2).

 DISCUSSION

Please discuss the effect sizes, expressed as partial Eta-squared (ηp2). Were the observed effects strong/mild/weak? Compare the effect sizes in your data with the effect sizes in previous studies.

 TO SUM UP I think the author(s) need to make the recommended corrections.

Author Response

Dear colleague, thanks for your review.

We inserted the preliminary data analysis specifying that all assumptions for statistical analyses (i.e., normality, homogeneity of variance, sphericity, multicollinearity, outliers, and independence of observations) were checked.
- We referred to Eta-squared partial (ηp 2 ); we just corrected in the text

- We also commented the effect in the discussion